

# Metagenomic analysis of orange colored protrusions from the muscle of Queen Conch *Lobatus gigas* (Linnaeus, 1758)

Jaison H. Cuartas[1], Juan F. Alzate[2], Claudia X. Moreno-Herrera[1] and Edna J. Marquez[1]

[1] Facultad de Ciencias, Laboratorio de Biología Molecular y Celular, Universidad Nacional de Colombia, Medellín, Antioquia, Colombia
[2] Facultad de Medicina, Centro Nacional de Secuenciación Genómica, Universidad de Antioquia, Medellín, Antioquia, Colombia

## ABSTRACT

The endangered marine gastropod, *Lobatus gigas,* is an important fishery resource in the Caribbean region. Microbiological and parasitological research of this species have been poorly addressed despite its role in ecological fitness, conservation status and prevention of potential pathogenic infections. This study identified taxonomic groups associated with orange colored protrusions in the muscle of queen conchs using histological analysis, 454 pyrosequencing, and a combination of PCR amplification and automated Sanger sequencing. The molecular approaches indicate that the etiological agent of the muscle protrusions is a parasite belonging to the subclass Digenea. Additionally, the scope of the molecular technique allowed the detection of bacterial and fungi clades in the assignment analysis. This is the first evidence of a digenean infection in the muscle of this valuable Caribbean resource.

## INTRODUCTION

The queen conch, *Lobatus gigas,* is an endangered marine gastropod of great socioeconomic, cultural and ecological importance in the Caribbean region. This species was included in Appendix II of the Convention on International Trade in Endangered Species of Wild Fauna and Flora (CITES) in 1992 and the Red List of the International Union for Conservation of Nature (IUCN) in 1994. Despite these regulations, natural stocks of this species continue to decline (*Theile, 2001*; *Aldana, 2003*), likely by the loss of breeding habitats and detrimental human activities such as overfishing (*Glazer & Quintero, 1998*; *Aldana, 2003*).

Compared with studies in basic biology (*Randall, 1964*), fisheries (*Brownell & Stevely, 1981*; *Theile, 2001*; *Prada et al., 2009*), and genetics (*Mitton, Berg Jr & Orr, 1989*; *Tello-Cetina, Rodríguez-Gil & Rodríguez-Romero, 2005*; *Zamora-Bustillos et al., 2011*; *Márquez et al., 2013*), parasitological and microbial studies of *L. gigas* are less explored (*Acosta et al., 2009*; *Aldana et al., 2011*; *Rodriguez, Hariharan & Nimrod, 2011*; *Pérez et al., 2014*). So far, only one parasitic infection, with *Apicomplexa* coccidian protozoon, has been reported in *L. gigas* (*Baqueiro et al., 2007*; *Aldana et al., 2009*; *Aldana et al., 2011*; *Gros, Frenkiel &*

Corresponding author
Edna J. Marquez,
ejmarque@unal.edu.co

*Aldana, 2009*; *Volland et al., 2010*). Similarly, only three published studies report the association between *L. gigas* and bacteria of the family Vibrionaceae (*Acosta et al., 2009*), the phyla Firmicutes, Proteobacteria, Actinobacteria (*Pérez et al., 2014*) as well as potential bacterial pathogens (*Rodriguez, Hariharan & Nimrod, 2011*). Two recent investigations have also studied the symbiotic association of *L. gigas* with dinoflagellates of the genus *Symbiodinium* (*Banaszak, Ramos & Goulet, 2013*; *García Ramos & Banaszak, 2014*).

Moreover, an unknown etiological agent sporadically produces orange colored protrusions in the muscle of *L. gigas* in the Colombian San Andres archipelago. However, it remains to be elucidated whether such lesions are caused by different agents and posteriorly colonized by pigment-producing microorganisms or digenean infections as found in other marine gastropods. Specifically, the infections of *Cercaria parvicaudata* and *Renicola roscovita* have been reported to produce orange/lemon colored sporocysts in different tissues of *Littorina* snails (*Stunkard, 1950*; *Galaktionov & Skirnisson, 2000*), whereas *Renicola thaidus* has been found infecting *Nucella lapillus* (*Galaktionov & Skirnisson, 2000*). These trematodes, *C. parvicaudata*, *R. roscovita* and *R. thaidus* are considered synonymous based on morphological similarities and cercariae size parameters (*Werding, 1969*). Similarly, lemon-cream to orange colored sporocysts are produced by the congeners *Renicola* sp. "polychaetophila" and *Renicola* sp. "martini" in infections of the gonad and digestive glands in *Cerithidea californica* (*Hechinger & Miura, 2014*).

This work studied the presence of parasites, bacteria and fungi in orange colored protrusions in the muscle of Colombian Caribbean queen conchs. This was achieved by using histological analysis and molecular approaches based on 454 FLX and capillary automated sequencing using an ABI PRISM 3100 Genetic Analyzer (Applied Biosystems, Foster City, CA, USA). This 454 FLX next-generation platform (Roche, Basel, Switzerland) permits high-throughput identification of hundreds of samples at reasonable cost and time consumption (*Mardis, 2008*). This approach allows functional analysis of sequencing data sets for comparative analysis of microbiome diversity of orange colored protrusions found in the muscle of *L. gigas* by using metagenomic taxonomical classifiers (*Huson et al., 2007*; *Huson et al., 2011*). This information is required for queen conch conservation and management strategies of potential pathogenic infections for human beings.

## MATERIALS AND METHODS

Orange colored protrusions were taken from three pieces of frozen muscle from one specimen of *L. gigas* processed for food trading in the Colombian Caribbean, San Andres archipelago (between 12°–16°N and 78°–82°W). These samples were provided by the Gobernación del Archipiélago de San Andrés, Providencia y Santa Catalina, through the scientific cooperation agreement #083/2012.

Since the etiological agent of these orange colored protrusions was unknown, we used three approaches to elucidate the origin of these lesions: (1) histological analysis, (2) 454 pyrosequencing of one whole genome shotgun library and (3) automated capillary sequencing (Sanger) of PCR amplified products to confirm the results provided by the metagenomic analysis. For histological analysis, samples from orange colored muscle

were fixed in 10% neutral phosphate-buffered formalin. The samples were prepared for histological examination by paraffin wax techniques and stained with hematoxylin and eosin following standard protocols (*García del Moral, 1993*; *Prophet et al., 1995*).

Due to scarcity of samples, the orange protrusions were pooled and ground with liquid nitrogen to extract the genomic DNA using the commercial DNAeasy Blood & Tissue Kit (Qiagen, Hilden, Germany), according to manufacturer recommendations. Sample pooling was performed to obtain high-quantity and high-quality DNA required for the generation of the genomic library. Purified DNA from the pooled sample was sequenced using the 454 Whole Genome Shotgun strategy according to standard protocols recommended by 454 GS FLX platform (Roche, Basel, Switzerland) at the Centro Nacional de Secuenciación Genómica, Universidad de Antioquia (*Margulies et al., 2005*). The obtained raw reads were end polished of low-quality regions with the toolkit PRINSEQ lite (*Schmieder & Edwards, 2011*) and assembled using MIRA3 v3.4 software (*Chevreux, Wetter & Suhai, 1999*).

Classification of assembled contigs was carried out using the BLAST algorithm against nucleotide and protein non-redundant databases of the NCBI with further computation of the taxonomic position of the assembled dataset with MEGAN software v5.5.3 (*Huson et al., 2011*). This metagenomic software uses a Lowest Common Ancestor-based algorithm that assigns each contig to taxa such that the taxonomical level of the assigned taxon reflects the level of conservation of the sequence (*Huson et al., 2007*). Then, species-specific and widely conserved sequences were assigned to particular taxa as described by *Huson et al. (2007)*. The contigs were classified using a bit-score threshold of 50, retaining only those hits that were within 10% of the best hit for a contig. Additionally, the *E*-value confidence criterion was set at 1E–15, even though a threshold value of 1E–04 is considered a good match (*De Wit et al., 2012*). Only contig alignment lengths above 100 nucleotides for BLASTN comparison or 100 amino acids for BLASTX comparisons were included in the assignment analysis. These analyses, comparing DNA or protein sequences, were carried out independently.

Furthermore, protein analysis assignments were classified to the proper taxonomic level according to *Monzoorul Haque et al. (2009)*, who empirically proposes identity thresholds for restricting the assignments. Alignments having identities in ranges of 61–65%, 56–60%, 51–55% and 41–50% were conservatively restricted in the level of family, order, class and phylum, respectively. The identity threshold of 66–100% was used for restricting the assignment of contigs to either species or genus or family levels. Additionally, the taxonomic level within this identity range was distinguished by the difference between the two alignment parameters, the percentage of identities and positives.

Moreover, a 1,000 bp fragment of the mitochondrial *cytochrome c oxidase I* gene was amplified by PCR following conditions reported by *Leung et al. (2009)* and primers described by *Bowles et al. (1993)* (JB3: 5′-TTTTTTGGGCATCCTGAGGTTTAT-3′) and *Králová-Hromadová (2008)* (trem.cox1.rrnl: 5′ AATCATGATGCAAAAGGTA-3′). This sequence was used instead of ribosomal genes since the bioinformatic analysis indicated a high enrichment of molluscan and some fungi ribosomal sequences, which limited the amplification of the helminth sequences of 18S and 28S ribosomal genes (data not shown). The *cytochrome c oxidase I* amplicon was sequenced by automated Sanger method using

an ABI PRISM 3100 Genetic Analyzer (Applied Biosystems, Foster City, CA, USA) and compared by BLASTN against the NCBI nucleotide database to look for sequence matches of reported organisms.

Finally, a Bayesian tree was constructed using the sequence obtained from orange colored protrusions and published sequences of Platyhelminthes. Bayesian tree construction was performed using MrBayes (MB) V3.2 (*Ronquist et al., 2012*) setting the GTR+I+G4 substitution model estimated by the software IQ-TREE, with 1,000,000 generations sampled every 1,000 generations and the other analysis parameters as default values. The convergence of the Markov Chain Monte Carlo iterations was assessed with the Potential Scale Reduction Factor (PSRF = 1; *Gelman & Rubin, 1992*) and the standard deviation of split frequencies (0.001).

# RESULTS

## Assembly and metagenomic approach

The massive shotgun sequencing generated 515,368 reads with an average length of 279 bp that were cleaned and then assembled using MIRA software into 5,180 contigs. Taxonomic classification of the contigs was carried out using the software MEGAN. For this analysis, all the contigs were compared with the NCBI's non-redundant protein database using the software BLASTX. With this strategy, 1,588 (30.7%) contigs were assigned to taxa (Bacteria: 412; Eukaryota: 1,157; other: 19), 866 (16.7%) were unassigned and 2,726 (52.6%) presented no hits. As expected, the Eukaryota group was dominant due to the origin of the sample. Furthermore, the group Gastropoda was frequently found in this analysis (186 hits), although many sequences remained unclassified due to the poor representation of these organisms in the public databases. Many bacterial sequences were also identified; 19 were assigned to the fungi group and 22 sequences were assigned to the Trematoda category. No viral or protozoa sequences were detected.

Following the MEGAN pipeline, with nucleotide comparisons using BLASTN and the nt/nr database, results were poorly classified. One contig was assigned to the root, 462 (8.9%) to particular taxa (Bacteria: 267; Eukaryota: 191; other: 4), 32 (0.6%) were unassigned and 4,685 (90.5%) had no hits. At the nucleotide level, most of the sequences were left unclassified. This reflects the lack of sequences in the databases of closely related organisms to the ones reported in this research.

## Bacteria and fungi associated with orange colored protrusions

The ranges for the confidence criterion represented by the *E*-value, similarity and identity for protein comparisons are shown in Table 1. Bacteria assignments included the class *Gammaproteobacteria* and the phylum *Firmicutes,* which includes the orders *Bacillales* and *Lactobacillales* (Fig. 1). Specifically, the class *Gammaproteobacteria* showed 322 assignments for *Psychrobacter*, exhibiting hits to several types of proteins with strains of *Psychrobacter* sp. (Fig. 1, Table 1). The identity criterion for *Psychrobacter* sp. ranged from 67% to 100% (Table 1). Similarly, nucleotide sequences showed hits for different genomic regions of *Psychrobacter* sp. strains and congeners, displaying identities ranging from 72% to 99% and alignment lengths from 104 bp to 1,061 bp (Table 2).

**Table 1** Diversity content in bacteria and fungi clades found in a pooled sample of orange colored protrusions from *L. gigas* muscle using translated contig sequences and the taxonomic classifier MEGAN.

| Contig | Organism | Gene | Ranges | | | | Assignment[a] |
| --- | --- | --- | --- | --- | --- | --- | --- |
| | | | *E*-value | Positives (%) | Identities (%) | Length (aa) | |
| 8 | *Psychrobacter* sp. | binding protein, kinase, transporter, adaptor, hypothetical proteins, membrane protein, glycosylase | 0.000E+00; 1.325E−30 | 96; 100 | 96; 100 | 113; 259 | Species |
| 11 | *Psychrobacter* sp. | dehydrogenase, hypothetical proteins, catalase, cytoplasmic protein, propionase, transferase, chaperone, deaminase, membrane protein | 0.000E+00; 4.451E−73 | 94; 100 | 90; 99 | 105; 306 | Genus |
| 2 | *Psychrobacter* sp. | channel protein, hypothetical protein | 1.189E−107; 4.390E−77 | 77; 83 | 67; 73 | 183; 242 | Family |
| 2 | *Carnobacterium jeotgali*[l] | replication initiator, phosphorylase | 5.120E−93; 4.347E−87 | 99; 100 | 99; 100 | 135; 138 | Species |
| 2 | *Carnobacterium* sp.[l] | hypothetical protein; integrase | 2.691E−127; 4.938E−58 | 100; 100 | 100; 100 | 109; 183 | Species |
| 1 | *Carnobacterium* sp.[l] | integrase | 1.940E−61 | 86 | 80 | 129 | Genus |
| 1 | *Carnobacterium* sp.[l] | hypothetical protein | 2.200E−69 | 94 | 86 | 158 | Family |
| 2 | *Lactobacillus jensenii*[l] | hypothetical protein | 2.022E−78; 2.550E−60 | 73; 85 | 68; 81 | 166; 169 | Genus |
| 1 | *Enterococcus faecalis*[l] | hypothetical protein | 3.700E−45 | 68 | 44 | 172 | Phylum |
| 7 | *Brochothrix thermosphacta*[b] | kinase, transcriptional regulator, transporter, ribosomal protein, reductase, hypothetical proteins | 3.514E−162; 9.715E−69 | 100; 100 | 100; 100 | 180; 248 | Species |
| 6 | *Brochothrix thermosphacta*[b] | dehydrogenase, hypothetical proteins, transposase, transferase | 1.918E−148; 1.431E−53 | 76; 100 | 71; 99 | 120; 243 | Genus |
| 1 | *Planococcus antarcticus*[b] | hypothetical protein | 2.650E−55 | 90 | 80 | 106 | Family |
| 1 | *Bacillus cytotoxicus*[b] | synthetase | 7.050E−164 | 83 | 71 | 223 | Family |
| 1 | *Lactococcus lactis* subsp. Lactis[l] | replication protein | 6.020E−91 | 81 | 62 | 164 | Family |
| 1 | *Staphylococcus aureus*[b] | hypothetical protein | 2.050E−32 | 71 | 64 | 104 | Family |
| 1 | *Fusarium oxysporum* | glutamine-rich protein | 1.064E−16 | 56 | 41 | 243 | Phylum |
| 1 | *Fusarium oxysporum* | glutamine-rich protein | 2.900E−16 | 85 | 82 | 243 | Genus |
| 1 | *Neurospora tetrasperma* | hypothetical protein | 1.426E−58 | 92 | 90 | 107 | Genus |

**Notes.**
[a]The assignments were classified to the taxonomic level according to *Monzoorul Haque et al. (2009)*, [b]*Bacillales*, [l]*Lactobacillales*.
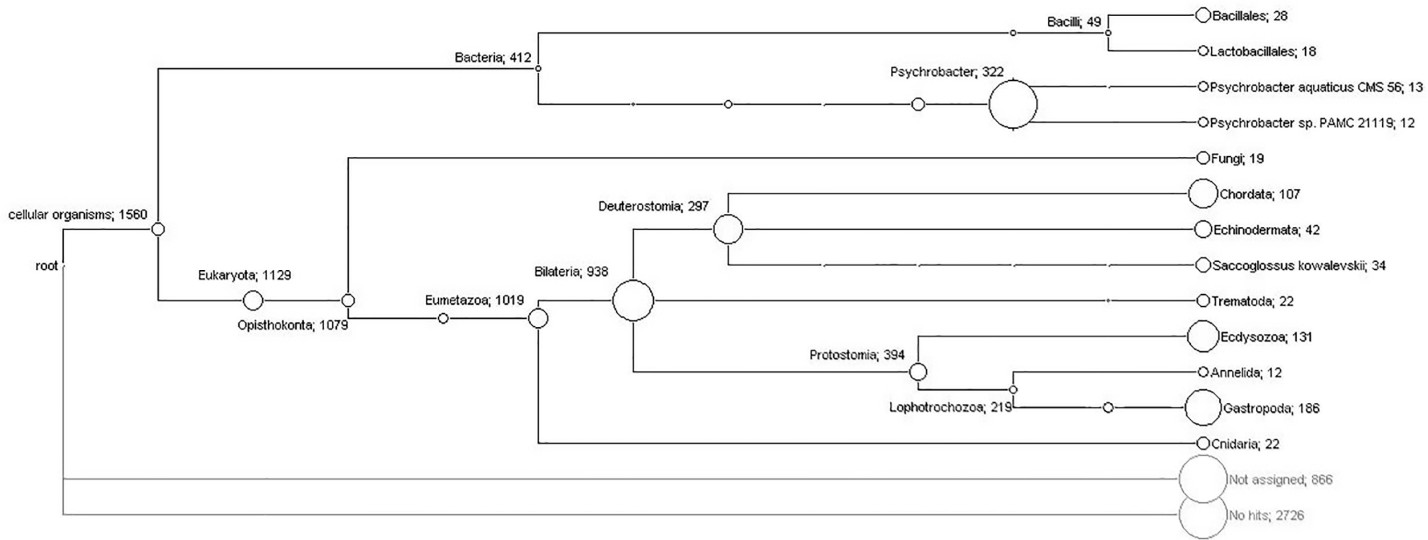

**Figure 1** **Phylogenetic diversity of translated contigs from orange colored protrusions of *Lobatus gigas* computed by MEGAN.** The nodes of the cladogram represent the assigned taxa and the numbers indicate the relative abundance of assigned contigs.

On the other hand, 18 assignments for the order *Lactobacillales* (Fig. 1) showed hits for diverse proteins exhibiting similarities and identities up to 100% for *Carnobacterium jeotgali* and *Carnobacterium* sp. (Table 1). We also found hits for proteins of *Lactobacillus jensenii* and *Enterococcus faecalis* displaying identities above 68% and 44%, respectively (Table 1). Similarly, nucleotide analysis showed matches for genome regions and plasmids of *Carnobacterium sp.*, displaying identities above 82% (Table 2). Additionally, the single hits for a plasmid and a genome fragment of *Enterococcus casseliflavus* and *Enterococcus faecalis* exhibited identities of 81% and 73%, respectively (Table 2).

A total of 28 contigs were assigned to different *Bacillales* bacteria within the phylum *Firmicutes* (Fig. 1); specifically, *Brochothrix thermosphacta* showed hits for several proteins exhibiting identities up to 100% (Table 1). *Planococcus antarcticus*, *Bacillus cytotoxicus*, *Lactococcus lactis* subsp. Lactis and *Staphylococcus aureus* showed identity ranges from 62% to 80% (Table 1). Furthermore, the nucleotide analysis showed hits for diverse bacteria belonging to genus *Listeria*, *Bacillus* and *Paenibacillus* (Table 2).

Only three out of 19 assignments to fungi clades satisfied the selection parameters; two hits supported the taxonomical levels of phylum and genus for *Fusarium oxysporum* and one hit classified to the genus taxonomical level for *Neurospora tetrasperma* (Table 1). In addition, nucleotide analysis (Table 2) showed three assignments for *Mrakia frigida* (rRNA genes, two hits) and *Togninia minima* (putative polyubiquitin protein mRNA, one hit).

## A parasite associated with orange colored protrusions

The histological approach showed a tissue lesion characterized by the aggregation of hemocytes (cells endowed with phagocytic and immune-related functions) inside isolated foci surrounded by smooth muscle fibers and a basophilic tissue contiguous to a lamellated

**Table 2** Diversity content in bacteria and fungi clades found in a pooled sample of orange colored protrusions from *L. gigas* muscle using nucleotide contig sequences and the taxonomic classifier MEGAN.

| Contig | Organism | Gene | *E*-value | Identities (%) | Length (bp) |
|---|---|---|---|---|---|
| | | | | **Ranges** | |
| 1 | *Psychrobacter* sp. | pRWF101 | 0.000E+00 | 98 | 993 |
| 1 | *Psychrobacter* sp. | gf | 0.000E+00 | 99 | 815 |
| 6 | *Psychrobacter* sp. | p, gf | 0.000E+00; 7.247E−59 | 90; 96 | 176; 679 |
| 1 | *Psychrobacter* sp. | gf | 0.000E+00 | 88 | 1,162 |
| 7 | *Psychrobacter* sp. | p, gf | 0.000E+00; 3.877E−131 | 80; 85 | 520; 914 |
| 3 | *Psychrobacter* sp. | gf | 4.648E−143; 2.430E−35 | 76; 79 | 288; 748 |
| 1 | *Psychrobacter cryohalolentis* | gf | 0.000E+00 | 93 | 1,047 |
| 3 | *Psychrobacter cryohalolentis* | p, gf | 5.595E−138; 6.961E−40 | 92; 95 | 112; 352 |
| 4 | *Psychrobacter cryohalolentis* | gf | 3.316E−153; 1.041E−103 | 83; 84 | 419; 537 |
| 3 | *Psychrobacter cryohalolentis* | gf | 1.626E−92; 3.574E−74 | 75; 78 | 524; 700 |
| 1 | *Psychrobacter arcticus* | gf | 0.000E+00 | 91 | 646 |
| 1 | *Psychrobacter arcticus* | gf | 5.090E−120 | 94 | 307 |
| 1 | *Psychrobacter arcticus* | gf | 8.830E−47 | 91 | 148 |
| 11 | *Psychrobacter arcticus* | gf | 0.000E+00; 3.900E−148 | 81; 88 | 611; 1,061 |
| 11 | *Psychrobacter arcticus* | gf | 0.000E+00; 6.195E−25 | 80; 89 | 104; 603 |
| 7 | *Psychrobacter arcticus* | gf | 1.962E−172; 1.181E−24 | 72; 79 | 224; 978 |
| 1 | *Carnobacterium* sp.[l] | gf | 0.000E+00 | 94 | 2,422 |
| 1 | *Carnobacterium* sp.[l] | pWNCR9 | 0.000E+00 | 92 | 1,466 |
| 1 | *Carnobacterium* sp.[l] | gf | 0.000E+00 | 96 | 1,244 |
| 1 | *Carnobacterium* sp.[l] | gf | 0.000E+00 | 98 | 1,029 |
| 3 | *Carnobacterium* sp.[l] | p, gf | 0.000E+00; 1.117E−155 | 95; 98 | 321; 624 |
| 6 | *Carnobacterium* sp.[l] | p, gf | 0.000E+00; 8.843E−57 | 82; 88 | 264; 897 |
| 1 | *Enterococcus casseliflavus*[l] | pTnpA | 4.680E−85 | 81 | 464 |
| 1 | *Enterococcus faecalis*[l] | gf | 3.310E−37 | 73 | 537 |
| 1 | *Listeria grayi*[b] | 23S rRNA | 0.00E+00 | 90 | 1,263 |
| 1 | *Listeria welshimeri*[b] | 23S rRNA | 1.150E−176 | 88 | 539 |
| 1 | *Listeria monocytogenes*[b] | gf | 3.850E−103 | 74 | 902 |
| 1 | *Listeria innocua*[b] | gf | 7.870E−109 | 76 | 754 |
| 1 | *Bacillus megaterium*[b] | gf | 5.410E−170 | 78 | 908 |
| 1 | *Bacillus toyonensis* [b] | gf | 1.430E−66 | 74 | 622 |
| 1 | *Bacillus cereus*[b] | gf | 2.570E−17 | 78 | 174 |
| 1 | *Paenibacillus larvae*[b] | pPL374 | 1.110E−170 | 100 | 335 |
| 1 | Uncultured compost bacterium[b] | 16S rRNA | 0.000E+00 | 99 | 436 |
| 1 | *Mrakia frigida* | 25S rRNA | 0.000E+00 | 100 | 1,429 |
| 1 | *Mrakia frigida* | 18S rRNA | 0.000E+00 | 99 | 1,793 |
| 1 | *Togninia minima* | protein mRNA | 3.407E−28 | 90 | 3,261 |

**Notes.**
[b] *Bacillales*, [l] *Lactobacillales*.
gf, genome fragment; p, plasmid; rRNA, ribosomal fragment.

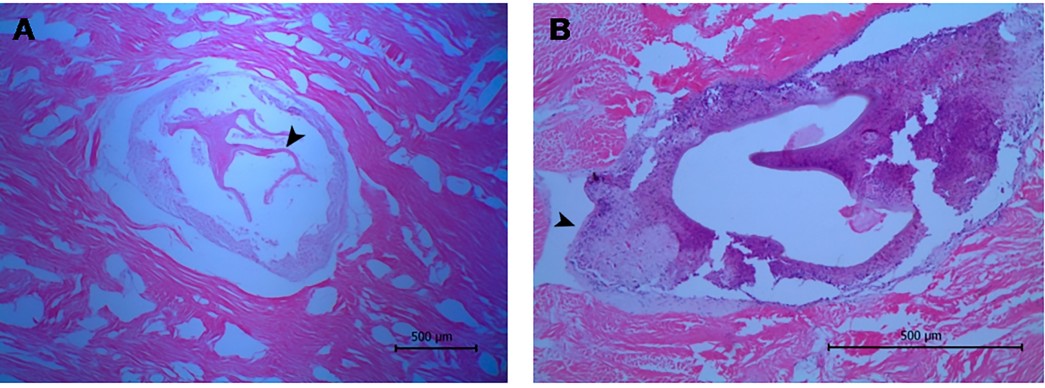

**Figure 2 Histological sections of orange-colored protrusions in the muscle of *Lobatus gigas*.** The lesions showed hemocytes stained purplish-blue and smooth muscle fibers pink-red in color. (A) Presence of lamellated membrane (arrowhead) (40×). (B) Granulation process (arrowhead) (100×).

membrane (Fig. 2A). Additionally, some lesions exhibit interstitial immunocyte inclusions with morphology similar to a granulation process (Fig. 2B). Although the histological approach did not allow for detectection of key features for identification, the microscopic images showed structures around 0.55 mm in diameter, which are compatible with immature developmental stages of a trematode.

Furthermore, the metagenomic analysis assigned 22 contigs to the trematode parasites clade. Specifically, these contigs had hits to an endonuclease-reverse transcriptase of *Schistosoma mansoni* (17) and *Schistosoma japonicum* (4), showing identities above 46% and 42%, respectively. Similarly, in the nucleotide analysis, seven contigs showed identities above 71% for different regions of two chromosomes of *S. mansoni* (Table 3).

We successfully amplified and sequenced a 740 bp region that confirmed the presence of trematode DNA in the *L. gigas* tissue (GenBank accession number KR092371). Moreover, this sequence clustered in a basal position to the suborder Xiphidiata (Trematoda: Digenea), which encompasses *Renicola* and *Helicometrina* genera (posterior probability: 0.98; Fig. 3). Additionally, the BLASTN results showed hits for some members of Xiphidiata, such as *Helicometrina labrisomid* (query coverage: 89%; identity: 77%), *Renicola cerithidicola* (query coverage: 70%; identity: 78%), *Synthesium pontoporiae* (query coverage: 42%; identity: 83%) and *Haematoloechus* sp. (query coverage: 39%; identity: 77%).

## DISCUSSION

In this study, three approaches, including histological analysis, 454 pyrosequencing and automated Sanger amplification of the *cytochrome c oxidase I* gene, were used to explore the potential causal agent of orange colored protrusions in the muscle of *L. gigas*. Identification by histology was limited since no characteristic structures were detected in the sample. Also, several contigs had no hits for proteins (∼52%) or nucleotide sequences (∼90%), indicating a lack of information on such sequences in reference databases. This explanation is plausible since the current protein sequence reference databases cover only a small
**Table 3** Diversity content in the trematoda clade found in a pooled sample of orange colored protrusions from *L. gigas* muscle using contig sequences and the taxonomic classifier MEGAN.

| Contig | Organism | Gene | Ranges | | | | Assignment [a] |
| | | | *E*-value | Positives (%) | Identities (%) | Length | |
|---|---|---|---|---|---|---|---|
| **Translated contig sequences** | | | | | | | |
| 2 | *Schistosoma japonicum* | endonuclease-reverse transcriptase | 3.637E–61; 6.062E–42 | 75; 76 | 64; 64 | 141; 165 | Family |
| 2 | *Schistosoma japonicum* | endonuclease-reverse transcriptase | 2.102E–122; 1.995E–56 | 61; 63 | 42; 43 | 262; 489 | Phylum |
| 5 | *Schistosoma mansoni* | endonuclease-reverse transcriptase | 3.147E–152; 4.075E–45 | 77; 81 | 61; 67 | 155; 345 | Family |
| 5 | *Schistosoma mansoni* | endonuclease-reverse transcriptase | 2.497E–172; 2.424E–61 | 71; 74 | 56; 59 | 204; 346 | Order |
| 4 | *Schistosoma mansoni* | endonuclease-reverse transcriptase | 0.000E+00; 9.203E–89 | 69; 70 | 52; 52 | 275; 695 | Class |
| 3 | *Schistosoma mansoni* | endonuclease-reverse transcriptase | 1.343E–66; 1.719E–51 | 65; 68 | 46; 50 | 193; 262 | Phylum |
| **Nucleotide contig sequences** | | | | | | | |
| 1 | *Schistosoma mansoni* | chromosome fragment W | 1.320E–20 | 80 | 80 | 161 | – |
| 5 | *Schistosoma mansoni* | chromosome fragments | 1.640E–55; 7.006E–27 | 71; 73 | 71; 73 | 649; 763 | – |
| 1 | *Schistosoma mansoni* | chromosome fragment 4 | 1.990E–19 | 77 | 77 | 199 | – |

**Notes.**

[a] The assignments were classified to the taxonomic level according to *Monzoorul Haque et al. (2009)*.

fraction of the biodiversity believed to be present in the environment (*Wu et al., 2009*). Despite these limitations, the alignment lengths of the contigs ($\geq 100$ nucleotides or amino acids) and the bit-scores (50) used in this research ensure a reasonable level of confidence in the taxonomic assignments (*Huson et al., 2007*).

## Bacteria and fungi associated with orange colored protrusions

The scope of the massive sequencing approach allowed the detection of some bacteria previously reported as microbiota associated with *L. gigas* (*Acosta et al., 2009*; *Pérez et al., 2014*), as well as new reports. For instance, *Psychrobacter* sp. was found in the *L. gigas* muscle in both nucleotide and protein analyses (Tables 1 and 2). This outcome corroborates previous studies that found *Psychrobacter* sp. in environmental (*Acosta et al., 2009*; *Pérez et al., 2014*) and tissue (*Pérez et al., 2014*) samples from *L. gigas*.

However, this study also found bacteria and fungi that have not been reported so far in *L. gigas*. Specifically, homologous protein and nucleotide sequences of species (e.g., *Carnobacterium jeotgali*), family and genus of *Carnobacterium sp.* were detected in the *L. gigas* muscle (Tables 1 and 2). *Carnobacterium* strains have been reported to inhabit live fish and a variety of seafood, dairy and meat (*Leisner et al., 2007*).

In addition, this research found homologous protein and nucleotide sequences of genus *Bacillus* and *Enterococcus* in the affected tissue of *L. gigas*. *Bacillus* species have ubiquitous distribution, inhabiting different environments such as soils, rocks, vegetation, foods and

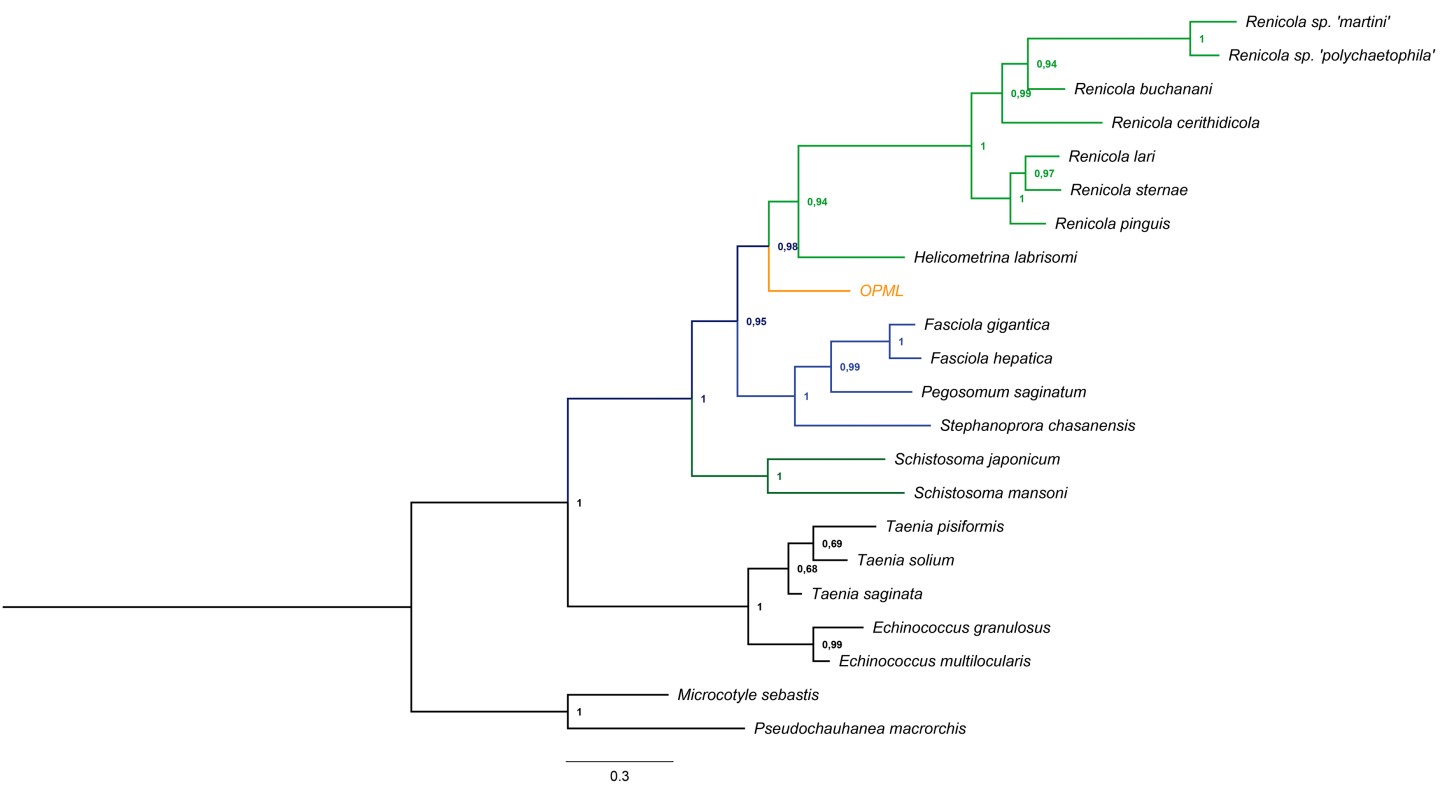

**Figure 3** **Bayesian tree obtained from *cytochrome c oxidase I* gene sequences of orange-colored protrusions from the muscle of *L. gigas* (OPML) and GenBank Platyhelminthes sequences.** *P. macrorchis* (JN592039.1), *M. sebastis* (NC_009055.1), *E. multilocularis* (AB018440.2), *E. granulosus* (AF297617.1), *T. saginata* (AY195858.1), *T. solium* (AY211880.1), *T. pisiformis* (GU569096.1), *S. mansoni* (AF216698.1), *S. japonicum* (AF215860.1), *S. chasanensis* (KU757308.1), *P. saginatum* (KX097855.1), *F. hepatica* (AF216697.1), *F. gigantica* (KF543342.1), *H. labrisomi* (KJ996009.1), *R. pinguis* (KU563724.1), *R. sternae* (KU563723.1), *R. lari* (KU563727.1), *R. cerithidicola* (KF512573.1), *R. buchanani* (KF512572.1), *Renicola* sp. 'polychaetophila' (KF512551.1) and *Renicola* sp. 'martini' (KF512560.1).

waters (*Nicholson, 2002*). Similarly, the ubiquitous nature of enterococci determines their frequently being found in foods as contaminants, although their predominant habitat is human and animal gastrointestinal tracts (*Giraffa, 2002*). However, they also occur in soil, surface waters, vegetables and fermented foods such as sausages, meat and cheese (*Giraffa, 2002*; *Foulquié et al., 2006*).

Furthermore, another bacteria present in the sample was *Brochothrix thermosphacta*, as it was assigned to bacterial species or genus taxonomical levels according to *Monzoorul Haque et al. (2009)*. This bacterium, closely related to *Listeria*, is a non-proteolytic food spoilage organism in prepacked meats and fish products (*Gardner, 1981*; *Lannelongue et al., 1982*; *Pin, García de Fernando & Ordóñez, 2002*). In addition, some *Listeria* hits were detected in the nucleotide analysis, although the identity values did not allow species identification. This result is concordant with studies that have isolated *Listeria* members from freshwater and marine environments (*Colburn et al., 1990*; *El Marrakchi, Boum'handi & Hamama, 2005*).

Metagenomic analysis also showed some fungi assignments related to *Fusarium, Neurospora, Togninia* and *Mrakia*. Both *Fusarium* and *Neurospora* exhibit wide distribution,

including humid tropical and subtropical marine environments (*Steele, 1967*; *Turner, Perkins & Fairfield, 2001*; *Babu et al., 2010*; *Summerell et al., 2010*; *Jebaraj et al., 2012*; *Saravanan & Sivakumar, 2013*; *Kumar, Gousia & Latha, 2015*). Specifically, some *Fusarium* species are associated with infections in crustaceans and cultivated fishes (*Hatai, 2012*), whereas other species are endosymbionts of some seaweeds (*Suryanarayanan, 2012*), corals (*Raghukumar & Ravindran, 2012*) and some sea sponges (*Höller et al., 2000*; *Wang, Li & Zhu, 2008*; *Liu et al., 2010*; *Paz et al., 2010*).

In contrast, *Togninia* and *Mrakia* show more restricted distributions. For instance, *Togninia* comprises pathogenic fungi responsible for the development of wood diseases, and some strains have been isolated from submerged wood from streams, lakes, ponds, reservoirs and ditches (*Hu, Cai & Hyde, 2012*). Likewise, several *Mrakia* species have been isolated from icy environments, including meltwaters from glaciers and permafrost in Antarctica (*Hua et al., 2010*; *Pathan et al., 2010*; *Carrasco et al., 2012*; *Zhang et al., 2012*; *Tsuji et al., 2013a*; *Tsuji et al., 2013b*), Argentina (*Brizzio et al., 2007*; *De Garcia, Brizzio & Broock, 2012*), the Qinghai-Tibet Plateau (*Su et al., 2016*), Italy (*Turchetti et al., 2008*; *Branda et al., 2010*; *Thomas-Hall et al., 2010*) and the Arctic (*Pathan et al., 2010*).

Considering that several of the new bacteria reports are related to food microorganisms, we hypothesized that they might grow under environmental or freezing conditions instead of being native microbiota. Fungi findings suggest an environmental source; however, since some species of *Fusarium* and *Neurospora* produce orange spores (*Davis & Perkins, 2002*; *Hatai, 2012*), the colored protrusions found in *L. gigas* may be due to an opportunistic or primary fungal infection. Thus, the role of bacteria and fungi in the muscle of *L. gigas* and their relationship with the lesion, native microbiota or the environment remains to be explored.

## A parasite associated with orange colored protrusions

Histology showed evidence of a membrane, that is consistent with a syncytium, enclosing a multicellular parasite, a mollusk inflammatory response elicited by hemocytes (*De Vico & Carella, 2012*). Moreover, such membranes are also compatible with the wall layers of the life cycle stage of Platyhelminthes, suggesting a possible infection by trematodes that infect other mollusks (*Cake, 1976*; *Sorensen & Minchella, 2001*). This finding was supported by the metagenomics analysis that showed sequences homologous to an endonuclease-reverse transcriptase of some species of trematodes like *Schistosoma* (Table 3). This result is expected since highly repetitive sequences, such non-LTR retrotransposons with an estimated copy number going up to 24,000, are more likely to be detected in whole genome shotgun amplification (*DeMarco et al., 2005*). Since databases of protein and nucleotide sequences are currently enriched with *Schistosoma*, but exhibit a poor representation of most members of Trematoda, these assignments require cautioned interpretation. Moreover, the lack of information in reference databases for most of the sequences from the studied sample (~90% of nucleotide sequences and ~52% of proteins) explains the relatively low number of hits for the parasite compared with Trematoda, bacteria and fungi taxa. Although these assignments are biased by the nucleotide and protein sequences available in the NCBI

databases, it supports the histological finding that the protrusions may be caused by a trematode.

The Bayesian tree supported the last result due to clustering of the sample in a basal position to the suborder Xiphidiata (Trematoda: Digenea), which includes *Renicola* species that produce colored pigments (*Stunkard, 1950*; *Galaktionov & Skirnisson, 2000*). The phylogenetic relationships with Xiphidiata were consistent with the BLASTN analysis that revealed genetic similitudes between the sequence found in this study and *Renicola, Helicometrina, Synthesium* and *Haematoloechus*, although its genetic distance with other members of these genera remains to be determined due to the lack of information for *cytochrome c oxidase I* and endonuclease-reverse transcriptase sequences of these taxa.

The molecular findings let us hypothesize about the structures approximately 0.55 mm in diameter found in the snail muscle tissue, although histology did not allow detection of key features for its identification. According to the life cycle described for Xiphidiata, the microscopic life cycle stage found in the muscle of *L. gigas* could be sporocysts, which are described to preferentially infect gonads and digestive glands, but can also disperse to other tissues in the form of more sporocysts or rediae (*Cribb et al., 2003*). Based on the structure's size, other stages such as metacercaria seems unlikely at least in *Renicola*, as they exhibit 0.12 to 0.16 mm in diameter (*Stunkard, 1964*). However, the evidence presented here is not enough to conclude which parasitic stage was observed within the colored lesions.

In conclusion, this study found evidence of a trematode infection, as well the presence of fungi and bacteria in the protruded muscle of *L. gigas*, which provides novel information for the parasitology and microbiology of this species. This first insight of a trematode infection in *L. gigas* is a baseline to expand the toolset to identify these organisms, the trematode life cycle, environmental conditions that trigger its appearance and epidemiological aspects regarding the host and possible effects on human health.

## ACKNOWLEDGEMENTS

The authors thank the Laboratorio de Patología Animal and Centro de Secuenciación Genómica, Universidad de Antioquia, for assistance and services provided. The authors also thank to the anonymous reviewers for their comments, which substantially improved the final version of this article.

### Funding

This work was funded by the Universidad Nacional de Colombia, Sede Medellín-the Gobernación del Archipiélago de San Andrés, Providencia y Santa Catalina, Grant 201010011420 (Scientific Cooperation Agreement # 083/2012), Convocatoria Nacional para el Programa Jóvenes Investigadores Colciencias No.525-2012-Identificación de parásitos asociados al caracol pala *Strombus gigas* en la Reserva de Biosfera Seaflower (Jaison H. Cuartas) and Programa de sostenibilidad 2016-2017, Grupo de Parasitología, Vicerrectoría de investigación, Universidad de Antioquia. The funders had no role in study design, data collection and analysis, decision to publish, or preparation of the manuscript.

## Grant Disclosures

The following grant information was disclosed by the authors:
Universidad Nacional de Colombia.
Sede Medellín-the Gobernación del Archipiélago de San Andrés.
Providencia y Santa Catalina: 201010011420.
Scientific Cooperation Agreement: #083/2012.
Convocatoria Nacional para el Programa Jóvenes Investigadores Colciencias No.525-2012-
Identificación de parásitos asociados al caracol pala *Strombus gigas* en la Reserva de Biosfera
Seaflower (Jaison H. Cuartas).
Programa de sostenibilidad, Vicerrectoría de investigación, Universidad de Antioquia.

## Competing Interests

The authors declare there are no competing interests.

## Author Contributions

- Jaison H. Cuartas, Juan F. Alzate, Claudia X. Moreno-Herrera and Edna J. Marquez conceived and designed the experiments, performed the experiments, analyzed the data, contributed reagents/materials/analysis tools, wrote the paper, prepared figures and/or tables, reviewed drafts of the paper.

## Ethics

The following information was supplied relating to ethical approvals (i.e., approving body and any reference numbers):

The muscle samples of *Lobatus gigas* were provided by the Gobernación del Archipiélago de San Andrés, Providencia y Santa Catalina, through the scientific cooperation agreement # 083/2012.

## DNA Deposition

The following information was supplied regarding the deposition of DNA sequences:
GenBank accession KR092371.

## Data Availability

The raw data has been supplied as a Supplemental File.

## Supplemental Information

Supplemental information for this article can be found online at http://dx.doi.org/10.7717/peerj.4307#supplemental-information.

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
