# Peer review of "Metagenomic analysis of orange colored protrusions from the muscle of Queen Conch Lobatus gigas (Linnaeus, 1758)"

_PeerJ, doi:10.7717/peerj.4307_

## Round 0.1 · original submission · Major Revisions

This manuscript requires major revision. The title of the study does not reflect the findings of it. The authors should clarify and refocus the story in the manuscript. There should be a clear connection between the title, objectives and findings/interpretations. Reviewers considered that the methodology used in this study was mostly adequate but needs re-focus.

Reviewer 1 ·

Basic reporting

Occasionally the writing is little hard to get through (although usually one can understand its intent). Some examples are:
L30 “explaining the nature of a plausible parasite infection” - awkward.
L50 overfishing would seem to be included in the detrimental human activities adduced later in the sentence.
L111 “lowest common ancestor” Unsure what this means.
L211 “taxonomical identification”… what other kind of identification is there?
L254 to say trematodes "have been reported" in mollusks reads like a strange sort of understatement (all digenetic trematodes infect mollusks).
L271-274 quite awkward.
L279-280 a reference is needed for the 40 million infections mentioned, as well as the “previous studies.”
L501-502. Fig 2 legend is awkwardly worded. Also, some of the nodes lack numbers, and circle size is not truly proportionate to hit numbers, as it seems the same for “no hits” (n=3000) and “not assigned” (n=900).
I am not an expert, but should this conch be referred to as Lobatus gigas?

Experimental design

The research gap (identity of the orange colored protrusion in conch) is well defined. However, I am puzzled by the methods chosen to address it, even if they are impressively multidisciplinary.

Few macroparasite infections can be reliably identified histologically (those that can are well studied taxa in well studied hosts). So, the use here of histology as primary identification method is surprising. It is also a relatively costly in terms of sample usage, which is a concern in a case like this, when there are only a few isolates in hand (L100).

As for pyrosequencing, the results speak for themselves. It does not seem surprising that the reads yield little information on the pathogen (the hits with bacteria may be of some interest, but I am not a good judge of that. However, I do find the reference to contamination for some of the more surprising bacterial hits a little understated at L249-250). Essentially, if I take a bottom-line look at the 454 results, they seem almost misleading, given the authors’ eventual conclusion: some are subject to possible contamination, and only a tiny minority are related to Trematoda.

Sanger sequencing could have been a potentially valuable approach, but by choosing CO1, the authors are hoping for a very specific match. Which would be great, but is a faint hope, given the largely unknown, unsequenced nature of most marine invertebrate taxa. For assignment to higher taxa – probably the most realistic, best-case outcome here – CO1 is a poor choice, because saturation of the third codon often makes deeper phylogenetic placement unreliable. For example, the authors cite Moszczynska et al, who recommend using ribosomal markers for higher taxonomic placement. 18S could well yield something more specific than Trematoda (even if it is not a great choice for identification at the species level). Moszczynska et al also reported sequencing individual metacercariae that were very small – on par with the specimens here. Pooling the specimens (L100) is unfortunate.

The analysis of the CO1 data is surprising. An NJ tree seems less appropriate that one based on character changes, such as ML, BI or MP.

It is disappointing that no specimens were examined in toto, stained, mounted on slides. This could have led to a more specific identification (in addition to being less work) and would have clarified the developmental stage of the trematodes. There is a reasonable chance these are metacercariae, not sporocysts or rediae. The authors seem to be assuming the latter at L275-277. However, it is metacercaraie that are responsible for foodborne trematodiases, which are discussed at several points.

Validity of the findings

The results appear valid, although as stated above, methods slightly different from those chosen might have yielded more satisfactory results (given the stated goal of the study).

At L291. Not very clear to me how any particular identification would yield any kind of practical preventative or control measure. Suggest elaborating with specifics or removing.

Additional comments

I don’t insist, but I feel like if the authors really want to identify this, they should sequence 18S and possibly ITS. Better chance of a more-informative near-match. Also, even if the data are left as is, NJ is not be best analytic method for the CO1 sequence.

Reviewer 2 ·

Basic reporting

The basic reporting was relatively clear; although, contextual and word usage corrections were made throughout the paper. I have made detailed corrections in my attached pdf of the article.
a) L26: Please replace ‘organisms’ with ‘taxonomic groups’ as no definitive species
identifications were made in this study.
b) L29: Please replace ‘class Trematoda’ with ‘subclass Digenea’.
c) L32: Please replace ‘meat’ with ‘muscle’.
d) L45: Please add ‘marine’ after endangered and remove ‘mollusk’.
e) L47-L49: The 2 conservation organizations that were cited are not named correctly.
Please rewrite with the correct names as follows: ‘Convention on
International Trade in Endangered Species of Wild Fauna and Flora’
‘Red List of the International Union for Conservation of Nature’
f) L56: Please replace ‘the parasitology and microbiology of this species’ with
‘;however, the parasitological and microbial studies of this species’.
g) L61: Please replace ‘phylum’ with ‘phyla’.
h) L66-69: Rewrite as follows: Moreover, the etiological agent of orange-colored
protrusions in the muscle of S. gigas remains unknown. These orange-
colored protrusions were observed sporadically in queen conchs from the
Columbian San Andres archipelago. This observation is characteristic with
digenean infection found in other marine gastropods.
i) L70: The phrase ‘orange/lemon-colored lesions’ is actually referring to the larval
digenean stage known as sporocysts. Replace the word ‘lesions’ to ‘sporocysts’.
j) L75: Please replace ‘lesions’ with ‘sprocysts’.
k) L80: What is the brand of the automated DNA sequencer used?
l) L82-84: Rewrite to clarify that the ‘marine samples’ are more specifically the
microbiome of orange-colored protrusions found in the muscle of S. gigas.

Experimental design

The experimental design was thorough with regards to the metagenomic analysis, however, this study was lacking in the morphological aspects of the digenean parasites and number of genes used for the genetic analysis with Sanger sequencing. Strombus gigas specimens could have easily been examined for the shedding of cercariae using non-destructive sampling methods by placing a live S. gigas specimen with orange-colored protrusions (sporocysts) in a container with purified seawater overnight. Morphological description of cercaria could then be compared to the other studies you cited (Stunkard, 1950; Galaltionov & Skirnisson, 2000; Werding , 1969; Hechinger & Miura, 2014). For gene sequencing a 740bp region of the mitochondrial COX1 gene was sequenced in this study. 28S rRNA gene sequence (AF023113) exist for Renicola
roscovita and other Renicola species. The ITS2 region would have also been helpful in
determining the taxonomic position of this unknown digenean species reported in the study.

a) L89: Please clarify whether orange-colored muscle is describing an anatomical
character of the queen conch or are you referring to the sporocysts. Where
these three samples of orange-colored “muscle” examined from a single
specimen or 3 different individuals?
b) L117-118: Please clarify if you intended to write ‘100 nucleotides or amino acids’. If
not, please replace with ’300 nucleotides or 100 amino acids’.
c) L132: What is the brand of the automated DNA sequencer used?
d) L150-151: Please explain why 22 sequencing assigned to the Trematode category
is remarkable, while 19 sequences assigned to the fungi group is
downplayed?
e) L254: It is a stretch to suggest cestode infection just because the histological
approach shows membrane consistent with tegument, especially since no
mention of cestode sequence came up in the metagenomic analysis. Please
remove ‘cestode’.
f) L263: Please replace ‘lesion’ with ‘protrusion’.
g) L523: I recommend that the cestode sequences should be removed from the
Neighbor Joining tree analysis and replaced with additional Digenea
sequence. The outgroup should be a specimen from the subclass
Aspidobothrea, which is the sister group of Digenea.

Validity of the findings

The validity of the findings with regards to evidence of Digenea infection of Strombus gigas could be much more robust with a non-destructive sampling method to recover shedding cercariae for the histological/ morphological analysis and use of additional nuclear gene sequencing if 18S, 28S, and ITS2 for the gene sequencing analysis.

Additional comments

I find the title of the article somewhat misleading. Based on the title of the article, I’m expecting a parasitological study that would provide solid morphological and molecular data to show evidence of digenean infection with data on larval stages (sporocysts) within and juvenile stages (cercaria) exiting Strombus gigas. Remarkably, roughly half of the results section and half of the discussion is dedicated to bacteria. This research seems to be more in line with a metagenomic study which is focused on the microbiome of “orange-colored protrusions” found in the muscle of Strombus gigas, then a study of digenean parasites infecting Strombus gigas.

Annotated reviews are not available for download in order to protect the identity of reviewers who chose to remain anonymous.

Reviewer 3 ·

Basic reporting

This study reports on the apparent presence of trematodes and bacteria in the tissues of Strombus gigas. Although the work is technically spectacular, the findings are so minor that I cannot recommend publication.
The approach used has demonstrated, probably convincingly, that the conch or conchs (not many infections and the samples were pooled) had a trematode infection. However, the molecular data are so generalised that there is no real identification beyond that of "trematode". This is my problem with the paper. Ultimately, probably every marine mollusc acts as intermediate host to trematodes at some stage. Without better identification or prevalence information we are really not significantly advanced by this study. Notably, the Discussion (Lines 275-6) states that the typical outcome of trematode infections in molluscs is partial or complete castration. The authors betray their lack of familiarity with these parasites here. Castration is associated with infection as a 1st intermediate host. The current infection is consistent with infection as a second int. host and likely to be relatively insignificant in the biology of the conch.
A great deal of the ms is devoted to issues associated with bacteria about which I know nothing. However, I do note that this makes the title of the paper rather misleading.
According to the WoRMS database, the name Strombus gigas is outdated - it should be Lobatus gigas; this does not inspire confidence.
In my view this study has not advanced knowledge of parasitism in this conch and I cannot therefor recommend publication

Experimental design

As above

Validity of the findings

As above

Additional comments

As above

---

## Round 0.2 · Minor Revisions

The reviewers were positive with the new resubmission. I encourage the authors to address the few minor modifications and submit a revised version.

Reviewer 1 ·

Basic reporting

The English writing has improved. It is still not fluent but there is no real barrier to understanding or communication.

I think the workflow used by the authors would be better represented, and their decisions more understandable, if they mention the specifics of their attempts to sequence 18S and 28S, as they do in the response (see paragraph above gel image in response), rather that making it sound like the attempt is a bad idea to begin with, as it comes across in the ms (L132-133). Also, the details of just how little tissue was available (only in the response) would be good to include in the MS proper.

Experimental design

I BLASTED the CO1 sequence and I think the fact that its highest similarity (82-83%) is to Synthesium, another member of the Xiphidiata, not mentioned in the MS because (presumably) it only overlaps at about 320 bp, is also worth mentioning in support of the eventual conclusion that the lesions are caused by a digenean belonging to the Xiphidiata.
Similarly, in addition to the statistical support of the clade membership, it would be relevant to state at L208-211 what ranges of similarities the sequences had with others in the tree.

I think the fact that 2 different translated contigs of endonuclease-reverse transcriptase indicate the specimens belong to the Schistosomatidae needs explanation. Either the method of Monzoorul Haque et al is flawed in this, or there is some other problem. The schistosomes are very evolutionary distant from the Xiphidiata.

Validity of the findings

I think the findings were, and remain valid.

I think the statement that the specimens are sporocysts (L202) (rather than metacercariae) requires justification. Some discussion of the known life history of related taxa might be pertinent.

Additional comments

Can the authors comment or speculate on why, if a digenetic trematode is what is causing these lesions, so many of the reads are from other taxa? If the sample's biomass is predominantly worm, than why are there few or no worm rDNA reads, but so many from fungi?

Reviewer 2 ·

Basic reporting

Since most of the corrections were made on the manuscript the basic reporting is very clear
and reads well. There is one correction that need to be made in the manuscript with the tracked changes as shown below.

a) L64-L67: This sentence needs to be rewritten as it is a run on sentence that
is awkwardly written.

Experimental design

Since most of the corrections were made on the manuscript the experimental design is more detailed and reads well. There are still a couple of corrections that either need to be made or were not addressed in my first review. The lines listed below are based on the manuscript with the tracked changes

a) L89-L90: Were these three samples of orange-colored "muscle" examined from a single
specimen or 3 different individual queen conch specimens?

b) L122-123: "Only contig alignment lengths above 100 nucleotides or 100 amino acids
were included in the assignment analysis." This still does not make sense
to me as 100 amino acids are equivalent to 300 nucleotides, but this is not
reflected in the text you have written in the text shown above.

Validity of the findings

The validity of the finding are now acceptable with regards to evidence of Digenean
infection of Strombus gigas as this is not the main focus of the paper as previously
written. This is especially true since changes were made to the title of the paper.

Additional comments

I'm happy to see that the authors took into to consideration the comments and corrections that
I previously made. The paper now reads much better and is clear and to the point. I believe that
this manuscript should be accepted with the minor revisions that I have listed in the above sections.

---

## Round 0.3 · accepted · Accept

The authors have addressed satisfactorily the comments and suggestions on the science aspect of this manuscript.

Reviewer 1 ·

Basic reporting

In the first review I mentioned that the writing had some problems. For the most part, these did not prevent comprehension but were distracting. I can see that the authors attempted to address this issue by making all the changes I specifically mentioned, but I’m afraid the problem remains. For publication purposes, a native speaker should be recruited to go over this closely. For what it's worth, I can sympathize: where I work the linguistic environment is not my first language, and often have to write/teach in a second language. ¡no es fácil!

Other than this, I have no major comments on the ms (one minor one, below), and I commend the authors on their work.

Experimental design

no comment

Validity of the findings

I think the authors could explicitly state the rationale behind their opinion about the developmental stage involved.